# Circulatory Agrin Serves as a Prognostic Indicator for Hepatocellular Carcinoma

**DOI:** 10.3390/cancers16152719

**Published:** 2024-07-31

**Authors:** Ankita Kapoor, Reza Bayat Mokhtari, Sahithi Savithri Sonti, Riya Patel, Anthony George, Kristopher Attwood, Renuka Iyer, Sayan Chakraborty

**Affiliations:** 1Department of Hematology-Oncology, Roswell Park Comprehensive Cancer Center, Buffalo, NY 14263, USA; ankita.kapoor@roswellpark.org (A.K.); sahithi.sonti@gmail.com (S.S.S.); riya.patel@roswellpark.org (R.P.); 2Department of Pharmacology and Therapeutics, Roswell Park Comprehensive Cancer Center, Buffalo, NY 14263, USA; reza.bayatmokhtari@roswellpark.org; 3Department of Biostatistics and Bioinformatics, Roswell Park Comprehensive Cancer Center, Buffalo, NY 14263, USA; anthony.george@roswellpark.org (A.G.); kristopher.attwood@roswellpark.org (K.A.); 4Program of Developmental Therapeutics, Roswell Park Comprehensive Cancer Center, Buffalo, NY 14263, USA

**Keywords:** agrin, hepatocellular carcinoma, prognosis, biomarker, extracellular matrix

## Abstract

**Simple Summary:**

There is an unmet need for the discovery of new biomarkers for detection and prognosis of liver cancers. Here we reveal the clinical potential of agrin as a key circulatory protein that predicts overall outcomes of liver cancer patients regardless of treatment regimens. Together, our study lays the foundation for further evaluation of agrin as a biomarker for improved clinical management of liver cancers.

**Abstract:**

Hepatocellular carcinoma (HCC), the predominant form of liver cancer, is associated with high mortality rates both in the United States and globally. Despite current advances in immunotherapy regimens, there is a scarcity of biomarkers to guide therapy selection. Alpha-fetoprotein (AFP) and glypican-3 have been proposed as biomarkers for HCC, but they do not provide any prognostic benefit for modeling disease progression. Agrin, a secreted proteoglycan, is frequently overexpressed in HCC and plays prominent role(s) in the liver tumor microenvironment (TME) to promote hepatocarcinogenesis. Here we employed a pilot single-center retrospective investigation to assess the prognostic value of agrin in HCC. Our evidence suggests that elevated serum agrin levels are associated with poor prognosis and performance among HCC patients. Multivariate Cox regression models indicate that secreted agrin serves as a better prognostic indicator compared to AFP that is significantly correlated with other secreted biomarkers (e.g., IL6). Cumulatively, this work demonstrates a promising clinical value of agrin in the detection and prognosis of HCC.

## 1. Introduction

Hepatocellular carcinoma (HCC) is the predominant form of primary liver cancer, accounting for over 90% of liver tumors [1]. According to the American Cancer Society, HCC ranks as the fifth most common cause of cancer-related mortality in males and the seventh most common in females in the United States [1,2]. HCC typically develops in individuals with chronic liver diseases, particularly cirrhosis, which introduces additional risks such as liver failure and contributes to the low 5-year survival rates ranging from 18% to 20% [3]. More than half of HCC-related deaths occur in Asia, primarily due to the prevalence of hepatitis B virus (HBV) infection. However, the incidence of HCC is rising in Western countries as well, driven by the increasing prevalence of non-alcoholic fatty liver disease, alcohol-related liver disease, and complications related to hepatitis C virus (HCV), even with the availability of direct-acting antiviral therapies [2,4,5]. Many patients with HCC are diagnosed at advanced stages and generally have limited treatment options, primarily palliative systemic therapies with suboptimal response rates, leading to 5-year survival rates of less than 5% [6,7]. Despite insights into the broad etiology of HCC, a better understanding of the complex molecular drivers of hepatocarcinogenesis is needed to leverage them as biomarkers and potential therapeutic targets [3].

Alpha-fetoprotein (AFP) has long served as the primary biomarker for HCC patients, playing a crucial role in early HCC screening, often in conjunction with imaging modalities like ultrasound or computed tomography [8]. Additionally, AFP is instrumental in postoperative surveillance for HCC recurrence. Its prognostic value in HCC has been well-established, with higher AFP levels often associated with early recurrence after definitive treatment, leading to diminished overall survival [4,9,10]. However, the prognostic utility of AFP remains a subject of debate in the HCC field. While some studies suggest that AFP levels may predict recurrence-free survival (RFS) following treatment, they do not necessarily correlate with overall survival (OS) [10,11]. Conversely, other studies have found that increased AFP levels independently predict poor OS [9]. Given the conflicting evidence surrounding AFP as a prognostic biomarker, there is a pressing need to identify additional biomarkers that can augment prognosis in HCC patients [5].

Agrin, a secreted heparan sulfate proteoglycan, has emerged as a promising candidate in this context. Agrin is frequently overexpressed and secreted in HCC and plays prominent roles in the liver tumor microenvironment (TME) to promote hepatocarcinogenesis and in the tumor mimicking wound healing and repair [12,13,14,15,16,17,18,19,20,21]. Our preclinical work has characterized agrin in regulating focal adhesion integrity, promoting oncogenesis in HCC models [18]. Agrin enhances cellular proliferation, migration, and oncogenic signaling by sustaining focal adhesion integrity and regulating extracellular matrix (ECM)–cancer cell communications [6,15]. Mechanistically, agrin’s extracellular matrix sensor activity provides oncogenic cues to regulate Arp2/3-dependent ruffling, invadopodia formation, and epithelial–mesenchymal transition through sustained focal adhesion integrity that drives liver tumorigenesis [19]. Furthermore, agrin signaling through the Lrp4-muscle-specific tyrosine kinase (MuSK) forms a critical oncogenic axis [7,19]. Importantly, antibodies targeting agrin have been shown to reduce oncogenic signaling and tumor growth in vivo, highlighting its potential as a therapeutic target [8,20,22].

Our previous studies demonstrated that agrin levels are significantly elevated in HCC tumor tissues and the serum of HCC patients, while low levels were observed in normal livers. Agrin expression was notably linked to tumor size and metastasis, suggesting its involvement in tumor growth and migration [12,20,23]. Additionally, agrin-positive HCC patients had a lower recurrence-free survival rate than agrin-negative patients, underscoring agrin’s potential as a prognostic marker [23,24]. These findings emphasize agrin’s significance in HCC progression and recurrence, warranting further investigation into its mechanistic role and therapeutic implications. To establish the clinical relevance of agrin as a prognostic biomarker in HCC, it is crucial to analyze its association with patient outcomes and other clinicopathological factors. Hence, we designed this study to gain clinical insights into the potential utility of serum agrin levels as a prognostic indicator by examining its correlation with survival outcomes, performance status, and other clinical characteristics in HCC patients [23,24]. In this retrospective study, we examined serum samples from patients with hepatobiliary cancers and explored the role of agrin and its relationship with patient outcomes.

## 2. Methods

### 2.1. Study Design and Patients

Our study was a retrospective, single-center investigation aimed at evaluating the role of agrin as a prognostic biomarker in hepatobiliary cancers. To mitigate the inherent limitations of a retrospective design, including potential biases related to patient selection and data collection, we employed a meticulous control matching process and applied rigorous statistical methods. These steps were taken to ensure the reliability of our findings, acknowledging that prospective studies are warranted to further validate our conclusions. The study included 89 patients (62 males, 27 females) newly diagnosed with various hepatobiliary cancers (such as HCC, gallbladder cancer, and intra- and extrahepatic cholangiocarcinoma) between February 2006 and September 2013. The diagnosis of hepatobiliary cancers was confirmed either histologically/cytologically or through hallmark findings evident on radiographic imaging.

A total of 90 controls were selected, matched for age, gender, and BMI, as these factors are known to significantly influence agrin levels. Blood samples collected from patients between February 2006 and September 2013 were subjected to blood tests. The serum concentration of agrin was measured using a commercially available enzyme-linked immunosorbent assay (ELISA) kit. Additionally, we aimed to assess the correlation between agrin levels and AFP, a well-established prognostic biomarker in hepatobiliary cancers. Although this study was conducted at a single center, the rigorous methodology and detailed data collection enhance the reliability of our findings. Future multicenter collaborations are planned to further validate these results across diverse populations.

### 2.2. Statistical Analysis

Patient demographic and clinical characteristics were summarized in the overall sample and by cohort (case versus control) using the mean, median, and standard deviation (SD) for continuous variables and frequencies and relative frequencies for categorical variables. Comparisons were made using GEE linear and multinomial regression models, as appropriate, accounting for within match correlation structure. All model assumptions were verified graphically, and age transformations were applied to the continuous outcomes. The distribution of agrin levels was displayed by cohort using dot plots, and the log-agrin levels compared using a GEE linear regression model.

In the subset of HCC patients, the agrin levels were summarized by levels of clinical and environmental factors using the mean, median, SD, and interquartile range (IQR). Associations were assessed using the Mann–Whitney U and Kruskal–Wallis exact tests, as appropriate. Overall survival was summarized by agrin quartiles using standard Kaplan–Meier methods where the median and 1-year rates were estimated with 95% confidence intervals, and comparisons were made using the log-rank test. A multivariable Cox regression model was used to evaluate the association between agrin levels and overall survival while adjusting for AFP (identified by stepwise selection). Hazard ratios and corresponding 95% confidence intervals were obtained from model estimates.

The correlations between agrin, AFP, and IL6 expression were evaluated using the Spearman correlation coefficient. The ability of these markers to differentiate between HCC (cases) and non-HCC (controls) was evaluated using conditional ROC curves (accounting for the within match correlation structure), from which the corresponding area under the curve (AUC) was obtained for each marker. Additionally, the Youden’s index criterion was used to identify “optimal” decision rules for each marker from which measures of marker performance (i.e., sensitivity, specificity, and positive/negative likelihood ratios) were obtained. As exploratory analyses, multivariable logistic regression models were considered to account for potential confounding factors (smoking, alcohol, and race (white versus non-white)).

All analyses are conducted in SAS v9.4 (Cary, NC, USA) at a significance level of 0.05.

## 3. Results

### 3.1. Baseline Characteristics

There was a total of 179 subjects in this study: 89 with newly diagnosed, unresectable hepatobiliary cancers and 90 healthy controls with no history of hepatobiliary cancers. The clinical characteristics of these subjects are listed in Table 1. Both cases and controls had the same proportion of males (68.7%) and females (31.3%), with appropriately matched age groups. A majority of the subjects in both cohorts were white (96.7% of those with cancers of interest and 85.4% of the control group). Within the newly diagnosed cancers there are 49 (55.1%) HCC cases: 8 (16%) stage I, 6 (12%) stage II, 16 (33%) stage III, 15 (31%) stage IV, and 4 (8%) not reported. The analysis of baseline characteristics revealed significant differences in agrin levels among various subgroups of patients. These differences highlight the potential influence of factors such as smoking status, alcohol consumption, and cirrhosis on agrin expression, underscoring the importance of considering these variables when evaluating agrin’s prognostic value [12,23,24].

### 3.2. Elevated Agrin Levels in Hepatobiliary Cancer Patients 

We evaluated the concentration of agrin levels in patients diagnosed with hepatobiliary cancer relative to matched controls from our biorepository, who exhibited no signs of cancer. As delineated in Table 1*,* individuals with hepatobiliary cancer demonstrated significantly higher agrin concentrations, averaging 7.97 ng/mL compared to 4.27 ng/mL observed in controls without a cancer history (Figure 1, *p* < 0.001).

### 3.3. Association of Clinical and Environmental Factors with Agrin Levels

To gain a more comprehensive understanding of the factors influencing agrin production, we investigated the association between serum agrin levels and the following variables: smoking status, alcohol consumption, viral hepatitis, cirrhosis, performance status, AFP levels, and other characteristics of malignancy, such as multifocality, venous thromboembolism, and portal vein involvement. These are listed in Table 2, and the most noteworthy associations are discussed below. 

(1)Performance status (ECOG)

A significant correlation was found between performance status (PS), as assessed by the Eastern Cooperative Oncology Group (ECOG) scale, and elevated agrin levels. Notably, patients with a PS greater than 2 demonstrated higher agrin concentrations, with a mean level of 10.5 ng/mL (*p* = 0.007). The strong correlation between elevated agrin levels and poorer performance status, as assessed by the ECOG scale, suggests that agrin may serve as a valuable indicator of disease progression and functional impairment in HCC patients [23,24,25]. This finding supports the potential clinical utility of agrin as a prognostic biomarker.

(2)Smoking

A notable difference in agrin levels was observed based on smoking history when comparing our total patient cohort. While agrin levels in healthy individuals remained stable regardless of smoking history, there was a significant increase in mean agrin levels among smokers compared to non-smokers within the HCC patient population. Specifically, smokers exhibited a mean agrin concentration of 9.49 ng/mL, whereas non-smokers had significantly lower mean agrin concentration of 6.8 ng/mL (see Table 2).

### 3.4. Survival Outcomes Predicted by Agrin

Lower concentrations of agrin exhibited a significant increased survival rate. Upon stratifying the range of agrin levels, patients with an agrin concentration between 4.9 and 6.3 ng/mL demonstrated a 63% one-year survival rate and a median survival duration of 16 months. In contrast, those with an agrin concentration exceeding 9.3 ng/ml had a markedly lower one-year survival of 14% and a significantly shorter median survival time of 3 months (*p* = 0.002), as delineated in Figure 2 and Table 3.

### 3.5. Survival Outcomes Comparing AFP and IL6 with Agrin

Multivariate Cox regression models demonstrated that an agrin concentration exceeding 9.3 ng/mL exhibited robust predictive capacity for survival outcomes (hazard ratio [9] 8.51, *p* < 0.001) when compared to elevated AFP levels (HR 1.55, *p* 0.25) as illustrated in Table 4. Furthermore, the presence of elevated AFP levels (>8 ng/mL) exhibited a strong positive association with increased agrin concentrations. Spearman’s rank correlation analysis revealed a significant positive correlation between high AFP levels and elevated agrin concentrations (correlation coefficient: 0.31944; *p* < 0.001), as depicted in Table 5. Likewise, compared to AFP, there was significant positive correlation between high IL6 and agrin levels (correlation coefficient: 0.43448; *p* < 0.001) (Table 5).

ROC analysis also revealed that agrin fared well as indicator of HCC but slightly below the level contributed by IL6 (ROC agrin = 0.92 vs. ROC IL6 = 0.97) (Figure 3). For agrin the optimal decision threshold was 4865.52 pg/mL (values above are classified as cases), which produces a sensitivity of 88.8%, specificity of 83.1%, and positive/negative likelihood ratios of 5.25 and 0.13, respectively. For IL6 the optimal decision threshold was 3.086 pg/mL (values above are classified as cases), which produces a sensitivity of 95.5%, specificity of 89.9%, and positive/negative likelihood ratios of 9.45 and 0.05, respectively. When combining the agrin and IL6, the model AUC is 0.96 and achieves a sensitivity of 91.0%, specificity of 91.1%, and positive/negative likelihood ratios of 10.2 and 0.10, respectively. In the multivariate analysis, adjusting for smoking status, alcohol use, and race (white versus non-white); both agrin and IL6 remain as significant predictors of HCC (*p* = 0.006 and *p* = 0.011, respectively). As such, higher levels of agrin and IL6 served as poor prognostic indicators for HCC with an HR of 2.57 (Table 5. Model 2 predictors). Considering the overlap of cases versus controls in secreted agrin levels below 10,000 pg/mL, the impact of agrin could be deduced as slightly lower than that of IL6 and biologically could be attributed to distinct cellular pathways initiated by these two secreted factors, suggestive of agrin as a complementing biomarker with IL6. 

As shown previously, our analysis revealed that serum agrin was significantly elevated in HCC patients who had a history of smoking or were current smokers (Table 6). Surprisingly, we found that agrin levels were lower in cirrhotic HCC patients and those who were exposed to alcohol intake when compared to controls and the entire patient cohort (Table 6).

## 4. Discussion

Our study aimed to evaluate agrin levels in patients with newly diagnosed, unresectable hepatobiliary cancers compared to healthy controls without a history of such cancers and to assess the associations between agrin levels and various factors, including smoking, alcohol consumption, viral hepatitis, cirrhosis, performance status, AFP levels, and other malignancy characteristics. The findings revealed that patients with hepatobiliary cancer exhibited significantly elevated agrin levels compared to healthy controls. Moreover, certain factors, such as performance status, smoking history, and elevated AFP levels, were strongly associated with higher agrin levels among cancer patients. As such, our data indicate that agrin may also complement well-known prognostic biomarkers such as IL6 for prediction of HCC.

From our study, elevated levels of agrin might be predictive of cirrhosis but may distinguish cirrhosis-induced HCC. Alcohol is one of the leading causes of liver diseases globally, resulting in a wide range of liver injuries including steatosis, alcoholic hepatitis, cirrhosis, and HCC. In our study, non-alcohol drinkers with HCC had higher agrin levels, probably indicating that other factors like genetics, viral exposure, and non-alcoholic steatohepatitis might be contributing to cirrhosis and thus HCC and might not be due to only alcohol exposure. Agrin levels alone might not distinguish alcohol to cirrhotic conditions, which could be due to a corrupt microenvironment and an interplay between different secreted extracellular matrix factors.

While our study provides significant evidence of agrin’s prognostic value, we acknowledge the need for further mechanistic studies to elucidate the role of agrin in HCC progression. Future research will focus on exploring the molecular pathways through which agrin exerts its effects on tumor growth and metastasis. While our study is indeed a single-center investigation, the comprehensive nature of our data collection and analysis provides a strong foundation for future multicenter studies. We also plan to collaborate with other institutions to validate our findings in a broader patient population.

AFP has been recognized as the principal biomarker for patients with HCC, characterized by a decline in levels following birth and maintenance of low concentrations throughout adulthood [7,26]. It assumes a pivotal role in the early screening of HCC, frequently utilized alongside imaging techniques such as ultrasound or computed tomography [8,27,28]. Furthermore, AFP is essential in the postoperative monitoring for recurrence of HCC. The prognostic significance of AFP in HCC is well-documented, where elevated levels of AFP are often correlated with early recurrence following definitive treatment, consequently leading to reduced overall survival [9,10,29,30,31].

However, the prognostic utility of AFP remains a subject of debate in the literature. While some studies, such as those by Schraiber Ldos et al. and Zhang et al., suggest that AFP levels may predict survival RFS following treatment, they do not necessarily correlate with OS [11,32]. Conversely, Silva et al. attempted to establish the prognostic significance of baseline serum AFP values in HCC patients and found that increased AFP levels independently predict poor OS [9]. Given the conflicting evidence surrounding AFP as a prognostic biomarker, there is a pressing need to identify additional biomarkers that can augment prognosis in HCC patients. These supplementary markers, such as agrin, might not only have the potential to aid in monitoring treatment response but also to guide the implementation of appropriate management strategies.

A significant association was observed between PS, as assessed by the Eastern Cooperative Oncology Group (ECOG) scale, and elevated agrin levels. Notably, patients with a PS greater than 2 exhibited higher agrin concentrations, with a mean value of 10.5 ng/mL. This finding suggests a potential correlation between deteriorating performance status and increased agrin expression, indicating a possible link between disease severity, functional impairment, and agrin secretion. It highlights the potential utility of agrin as a biomarker reflective of disease progression and functional impairment. The association between smoking and elevated agrin levels among patients diagnosed with hepatobiliary cancer is noteworthy. Smokers exhibited higher mean agrin concentrations compared to non-smokers, implying a potential role of smoking in modulating agrin expression within this patient population. This finding underscores the importance of considering lifestyle factors in comprehending cancer biomarkers and their implications for disease progression [33,34,35].

Additionally, elevated levels of AFP exhibited a positive correlation with increased levels of agrin, suggesting a potential interplay between these two biomarkers in hepatobiliary cancer. This correlation between agrin and AFP further underscores the complexity of the tumor microenvironment and its impact on biomarker expression. The positive correlation between elevated agrin and AFP levels, coupled with the superior prognostic capability of agrin in predicting survival outcomes, as demonstrated by the multivariate Cox regression analysis, highlights the potential of agrin as a more reliable prognostic biomarker than AFP in HCC. These findings underscore the need for further exploration of agrin’s role in disease monitoring and treatment stratification. Moreover, survival analysis revealed a significant association between lower levels of agrin and enhanced survival rates among patients with hepatobiliary cancer. Patients with reduced concentrations of agrin demonstrated higher one-year survival rates and extended median survival times compared to those with elevated levels of agrin. Multivariate Cox regression analysis confirmed the concentration of agrin as a robust predictor of survival outcomes, emphasizing its potential clinical utility as a prognostic biomarker in hepatobiliary cancer.

Zhang et al. conducted a study which demonstrated significantly higher agrin expression in HCC samples compared to normal tissue controls (77.1% vs. 25.4%, *p* < 0.05), indicating a potential role in HCC pathogenesis. Agrin expression exhibited a significant association with tumor size (*p* = 0.041) and metastasis (*p* = 0.034), suggesting its involvement in tumor growth and dissemination. Notably, HCC patients with agrin-positive tumors exhibited a lower recurrence-free survival rate compared to those with agrin-negative tumors (*p* = 0.001), highlighting the potential of agrin as a prognostic marker. These findings underscore the significance of agrin in HCC progression and recurrence, warranting further investigation into its mechanistic role and therapeutic implications [23]. To the best of our knowledge, no previous study has investigated serum agrin expression in HCC patients. Our study fills this gap, offering a novel approach that is more feasible due to the ease of serum measurement. Therefore, our results suggest that agrin has the potential to be used as a prognostic indicator, outperforming AFP.

The findings from this study unveil a crucial role of the proteoglycan agrin in the progression and prognosis of HCC. Agrin, a secreted heparan sulfate proteoglycan, was found to be significantly overexpressed and secreted in HCC tumor tissues and serum samples compared to normal liver tissues. This observation aligns with previous studies that have reported elevated agrin levels in various cancers, including hepatocellular carcinoma, oral squamous cell carcinoma, and cholangiocarcinoma [20,36,37].

Corroborating our findings, Zhang et al. reported that agrin expression was associated with tumor size, metastasis, and lower recurrence-free survival rates in HCC patients, further emphasizing agrin’s significance in HCC progression and recurrence [23]. Additionally, previous studies demonstrated that agrin silencing interfered with cancer cell motility, proliferation, invasion, colony formation, and tumor spheroid formation in oral squamous cell carcinoma, highlighting agrin’s pathological role in cancer progression [37,38,39]. Hence, our study will serve as a platform to analyze the role of secreted agrin as a biomarker in other solid cancers where agrin is mechanistically involved in oncogenic signaling.

While we acknowledge the retrospective nature of our study may introduce certain biases, we have endeavored to minimize these through careful control matching and the application of rigorous statistical analyses. We recognize the value of prospective studies in confirming our findings and recommend them as an essential next step for future research in this area, particularly testing in patients of diverse HCC origins. While our study is limited to a single center, the comprehensive nature of our data collection and analysis provides a strong foundation for future multicenter studies. We plan to collaborate with other institutions to validate our findings in a broader patient population, thereby enhancing the application of our results.

From a therapeutic perspective, our previous work has demonstrated that antibodies targeting agrin reduced oncogenic signaling and tumor growth in vivo, highlighting agrin’s potential as a therapeutic target for antibody therapy in HCC. Furthermore, the increased plasma agrin levels observed in HCC patients may also be exploited to develop diagnostic strategies for early detection and monitoring of the disease. 

In conclusion, our study provides compelling evidence for the oncogenic role of agrin in HCC progression and its potential as a prognostic biomarker in addition to being an attractive therapeutic target. The findings contribute to a deeper understanding of the molecular drivers of hepatocarcinogenesis and pave the way for further investigation into the mechanistic roles of agrin and its potential clinical applications in HCC management.

## Figures and Tables

**Figure 1 cancers-16-02719-f001:**
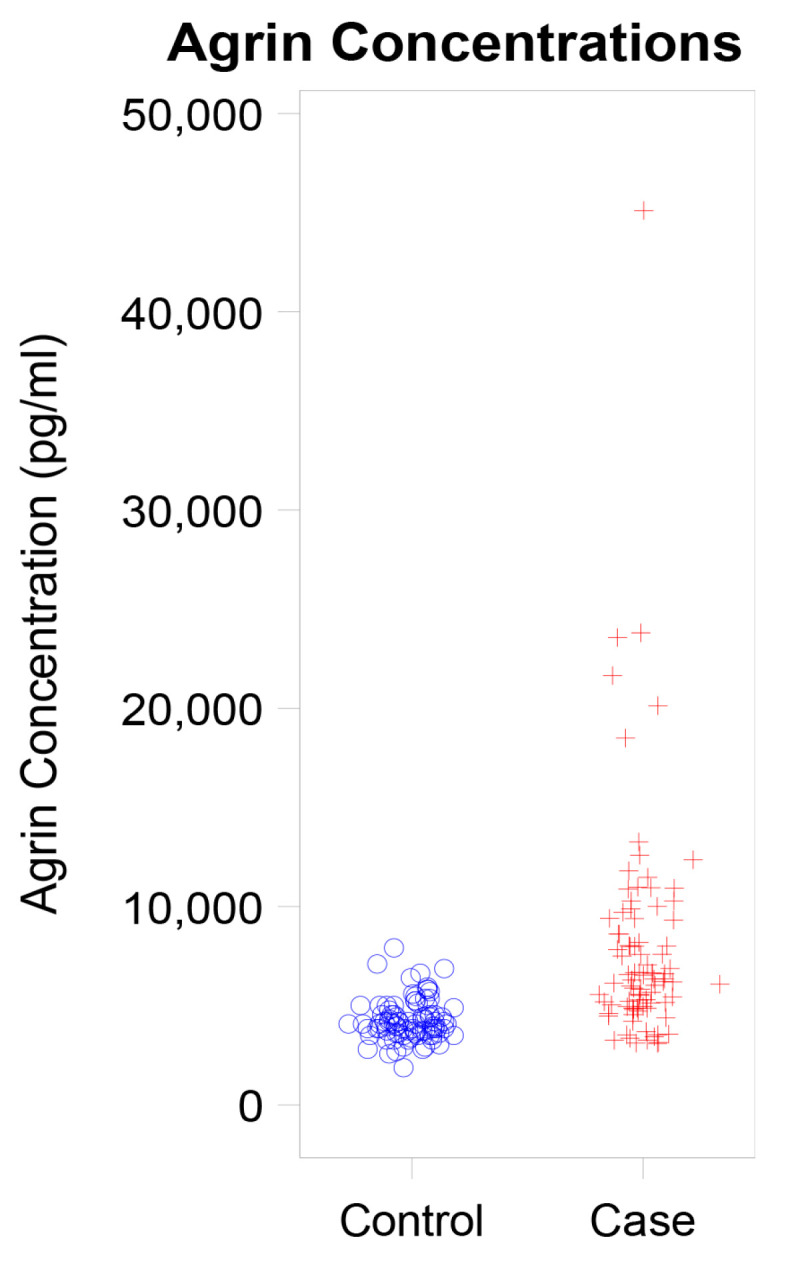
Dot Plot Showing Agrin Concentration in Cases versus Controls.

**Figure 2 cancers-16-02719-f002:**
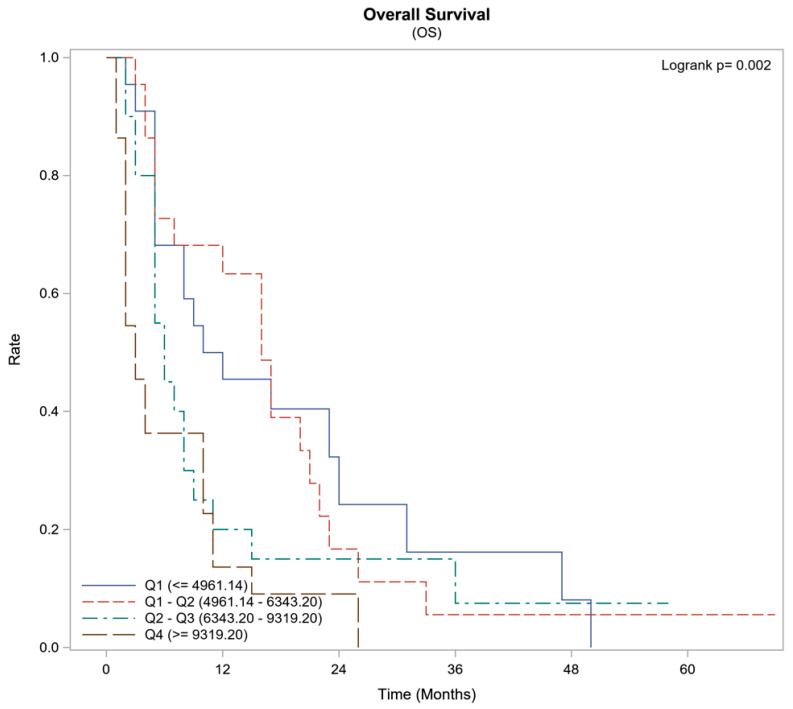
Survival Summary by Agrin Quartiles (Subset: HCC Patients Only).

**Figure 3 cancers-16-02719-f003:**
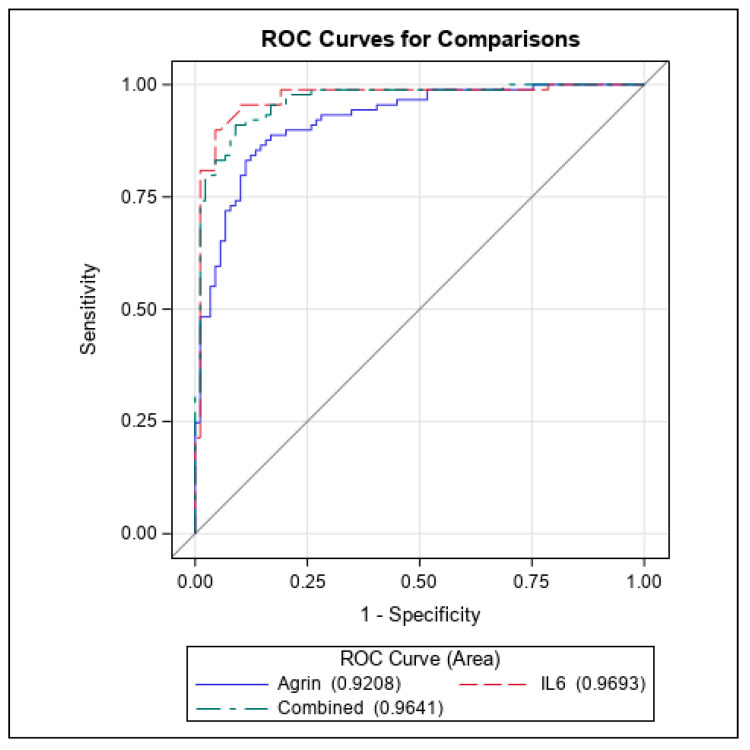
ROC curves of Agrin and IL6 as prognostic indicators for HCC.

**Table 1 cancers-16-02719-t001:** Summary of Patient Demographic Characteristics in Cases versus Controls.

		Control	Case	Overall	*p*-Value
	N	90 (50.3)	89 (49.7)	179 (100%)	
Age	Mean/Std/N	60.34/12.77/90	60.37/12.50/89	60.36/12.60/179	0.161
	Median/Min/Max	59.00/23.00/85.00	59.00/24.00/86.00	59.00/23.00/86.00	
Gender	Male	61 (67.8%)	62 (69.7%)	123 (68.7%)	0.162
	Female	29 (32.2%)	27 (30.3%)	56 (31.3%)	
BMI	Mean/Std/N	28.70/5.51/90	28.91/6.04/89	28.81/5.76/179	0.797
	Median/Min/Max	27.43/18.24/48.82	28.35/17.35/49.72	27.70/17.35/49.72	
Race	White	87 (96.7%)	76 (85.4%)	163 (91.1%)	0.003
	Black	3 (3.3%)	8 (9.0%)	11 (6.1%)	
	Asian		1 (1.1%)	1 (0.6%)	
	Native American		2 (2.2%)	2 (1.1%)	
	Multiracial		1 (1.1%)	1 (0.6%)	
	Other		1 (1.1%)	1 (0.6%)	
Smoking	Never	53 (59.6%)	27 (30.3%)	80 (44.9%)	0.007
	Current	5 (5.6%)	29 (32.6%)	34 (19.1%)	
	Former	31 (34.8%)	33 (37.1%)	64 (36.0%)	
Alcohol	No	14 (15.6%)	24 (27.0%)	38 (21.2%)	0.024
	Current	76 (84.4%)	35 (39.3%)	111 (62.0%)	
	Former		30 (33.7%)	30 (16.8%)	
Cirrhosis	No	86 (100.0%)	62 (69.7%)	148 (84.6%)	<0.001
	Yes		27 (30.3%)	27 (15.4%)	
HBV	No		84 (94.4%)	84 (94.4%)	
	Yes		5 (5.6%)	5 (5.6%)	
HCV	No		62 (69.7%)	62 (69.7%)	
	Yes		27 (30.3%)	27 (30.3%)	
Cholangitis	No		89 (100.0%)	89 (100.0%)	
Type of Cancer	Liver and Bile Ducts		79 (88.8%)	79 (88.8%)	
	Gallbladder		8 (9.0%)	8 (9.0%)	
	Other Biliary Tract		2 (2.2%)	2 (2.2%)	
AJCC Stage	I		11 (13.3%)	11 (13.3%)	
	II		9 (10.8%)	9 (10.8%)	
	III		24 (28.9%)	24 (28.9%)	
	IV		39 (47.0%)	39 (47.0%)	
ECOG	0		28 (31.5%)	28 (31.5%)	
	1		41 (46.1%)	41 (46.1%)	
	2+		20 (22.5%)	20 (22.5%)	
Uncontrolled Pain at Presentation	No		34 (38.2%)	34 (38.2%)	
	Yes		55 (61.8%)	55 (61.8%)	
AFP	Mean/Std/N	././0	15,065.88/44,978.87/59	15,065.88/44,978.87/59	
	Median/Min/Max	././.	181.70/1.30/307,716.6	181.70/1.30/307,716.6	
Multifocal Cancer on CT	No		32 (36.0%)	32 (36.0%)	
	Yes		57 (64.0%)	57 (64.0%)	
Portal Vein Thrombosis on CT	No		59 (66.3%)	59 (66.3%)	
	Yes		30 (33.7%)	30 (33.7%)	
VTE at Time of Diagnosis	No		79 (88.8%)	79 (88.8%)	
	Yes		10 (11.2%)	10 (11.2%)	
Agrin Level	Mean/Std/N	4.27/1.03/90	7.97/5.81/89	6.11/4.54/179	<0.001
	Median/Min/Max	4.12/1.87/7.90	6.34/3.10/4.51	4.90/1.87/45.10	

**Table 2 cancers-16-02719-t002:** Summary of Agrin Levels by Patient Demographic Characteristics (Subset: HCC Patients Only).

		*n*	Mean (SD)	Median (IQR)	*p*-Value
**Smoking**	Never	27	6861.3 (3961.3)	5835.4 (4878.4–8189.7)	0.32
	Current	29	9490.7 (8433.5)	6669.2 (5288.4–10,006.4)	
	Former	33	7541.8 (3841.1)	6545.2 (4871.2–9406.4)	
Alcohol	No	24	9676.0 (8774.8)	6849.8 (5702.9–9860.4)	0.39
	Current	35	7602.6 (4738.4)	5890.3 (4878.4–9391.2)	
	Former	30	7035.1 (3438.1)	6278.9 (4724.4–8189.7)	
Cirrhosis	No	62	8224.4 (6808.3)	6055.9 (4871.2–9391.2)	0.25
	Yes	27	7387.2 (2297.2)	6867.2 (5557.2–9319.2)	
HBV	No	84	8083.5 (5953.3)	6344.2 (4999.0–9398.8)	0.49
	Yes	5	6071.0 (2072.1)	5835.4 (4878.4–7587.8)	
HCV	No	62	8086.6 (6462.3)	6112.5 (4878.4–9391.2)	0.55
	Yes	27	7703.6 (4056.0)	6545.2 (5197.0–8616.4)	
Cholangitis	No	89	7970.4 (5817.2)	6343.2 (4961.1–9319.2)	
ECOG	0	28	6137.0 (2336.1)	6010.2 (4021.7–7793.8)	0.007
	1	41	7984.0 (7004.6)	6082.7 (5197.0–7989.7)	
	2+	20	10,509.2 (5826.1)	10,384.2 (6419.4–12,198.2)	
Uncontrolled Pain at Presentation	No	34	7674.2 (7443.8)	5959.0 (4547.8–7836.4)	0.09
	Yes	55	8153.5 (4603.8)	6646.5 (5197.0–10,006.4)	
AFP	Normal (≤8)	13	5546.9 (1457.2)	5890.3 (4724.4–6212.6)	0.03
	Elevated (>8)	46	8705.8 (6900.8)	6590.4 (5231.6–9406.4)	
Multifocal Cancer on CT	No	32	8347.9 (7470.8)	6254.5 (5152.1–9355.2)	0.96
	Yes	57	7758.5 (4706.7)	6345.2 (4949.2–8616.4)	
Portal Vein Thrombosis on CT	No	59	8323.6 (6887.2)	6202.4 (4865.5–9319.2)	0.50
	Yes	30	7275.8 (2658.2)	6647.4 (5288.4–9406.4)	
VTE at Time of Diagnosis	No	79	7819.3 (5954.1)	6212.6 (4878.4–8614.4)	0.13
	Yes	10	9164.3 (4674.9)	8388.8 (5673.1–10,286.4)	

**Table 3 cancers-16-02719-t003:** Survival outcomes based on Agrin Quartiles.

	1-yr Surv. Rate (95% CI)	Median Surv. (95% CI)	Sample	Log Rank *p*-Value
Total	0.36 (0.26, 0.46)	8.0 (5.0, 11.0)	E = 77 C = 9 T = 86	*p* = 0.002
Q1 (≤4961.14)	0.45 (0.24, 0.64)	11.0 (5.0, 24.0)	E = 18 C = 4 T = 22	
Q1–Q2 (4961.14–6343.20)	0.63 (0.40, 0.80)	16.0 (5.0, 21.0)	E = 19 C = 3 T = 22	
Q2–Q3 (6343.20–9319.20)	0.20 (0.06, 0.39)	6.0 (5.0, 9.0)	E = 18 C = 2 T = 20	
Q4 (≥9319.20)	0.14 (0.03, 0.31)	3.0 (2.0, 10.0)	E = 22 C = 0 T = 22	

**Table 4 cancers-16-02719-t004:** Univariate and Multivariate Cox Regression Modeling (Subset: HCC Patients Only).

**Univariate Cox Regression Models**		
**Agrin**	**Hazard Ratio (95% CL)**	***p*-Value**
Q1 (≤4961.14)	Ref.	0.0064
Q1–Q2 (4961.14–6343.20)	1.022 (0.536–1.951)	0.9468
Q2–Q3 (6343.20–9319.20)	1.530 (0.792–2.953)	0.2053
Q4 (≥9319.20)	2.659 (1.409–5.019)	0.0026
**Multivariate Cox Regression Models**		
**Model 1 Predictors: Agrin and AFP (Elevated vs. Normal)**		
**Agrin**	**Hazard Ratio (95% CL)**	***p*-Value**
Q1 (≤4961.14)	Ref.	<0.0001
Q1–Q2 (4961.14–6343.20)	1.094 (0.487–2.456)	0.8284
Q2–Q3 (6343.20–9319.20)	2.067 (0.871–4.903)	0.0997
Q4 (≥9319.20)	8.157 (3.085–21.567)	<0.0001
**AFP**	**Hazard Ratio (95% CL)**	***p*-Value**
Normal (≤8)	Ref.	0.2534
Elevated (>8)	1.553 (0.730–3.306)	

**Table 5 cancers-16-02719-t005:** Spearman Correlation Coefficients and multivariate models between Agrin and AFP/IL6.

**Spearman Correlation Coefficients** **Prob > |r| under H0: Rho = 0** **Number of Observations**			
	**Agrin**	**AFP**	**IL6**
**Agrin**	1.00000	0.31944	0.43448
		0.0137	<0.0001
	89	59	89
**AFP**	0.31944	1.00000	0.19852
	0.0137		0.1317
	59	59	59
**Multivariate Cox Regression Models**			
**Model 2 Predictors: Agrin and IL6 (Dichotomized at Median)**			
**Agrin**	**Hazard Ratio (95% CL)**		** *p* ** **-Value**
Q1 (≤4961.14)	Ref.		0.0198
Q1–Q2 (4961.14–6343.20)	1.005 (0.514–1.965)		0.9881
Q2–Q3 (6343.20–9319.20)	1.493 (0.738–3.020)		0.2645
Q4 (≥9319.20)	2.576 (1.258–5.275)		0.0097
**IL6**	**Hazard Ratio (95% CL)**		** *p* ** **-Value**
≤Median (17.199)	Ref.		0.8509
>Median (17.199)	1.050 (0.629–1.752)		

**Table 6 cancers-16-02719-t006:** Evaluation of serum agrin levels with smoking, alcohol, and cirrhosis conditions amongst all patients versus controls.

Patient Characteristic		Cases		Controls		*p*-Value
		N	Agrin Level Mean (SD)	N	Agrin Level Mean (SD)	
Smoking	Never	27	6861.3 (3961.3)	53	4174.9 (969.4)	0.0001
	Current	29	9490.7 (8433.5)	5	4115.2 (694.1)	0.0048
	Former	33	7541.8 (3841.1)	31	4518.1 (1154.3)	<0.0001
Alcohol	No	24	9676.0 (8774.8)	14	4795.4 (894.4)	0.0008
	Current	35	7602.6 (4738.4)	76	4182.1 (1032.0)	<0.0001
	Former	30	7035.1 (3438.1)			
Cirrhosis	No	62	8224.4 (6808.3)	86	4315.6 (1030.8)	<0.0001
	Yes	27	7387.2 (2297.2)			

## Data Availability

All data are available in the manuscript.

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
