# Peer review of "Circulatory Agrin Serves as a Prognostic Indicator for Hepatocellular Carcinoma"

_cancers, 2024, doi:10.3390/cancers16152719_

Round 1

Reviewer 1 Report

Comments and Suggestions for Authors

Congratulations on this paper. I have, although, some comments that I would like to address:

1. I seem to be confused and apologize for this, but you state that were included in this retrospective 89 subjects with advanced hepatobiliary cancer, but then you present in table 1 that almost a quarter of those have actually AJCC stage 1 and 2 cancers, being the remaining patients stage III and IV cancers. So, which one is it, are they all advanced hepatobiliary cancer patients or not?

2. You also note that the 89 patients have various types of hepatobiliary cancer, with only 88.9% of them having liver & bile ducts cancer. I assume that a fraction of those patients have HCC. But then, you state in table 2 that you included a subset of HCC patients, which sum… 89 in total. I would like to know exactly, how many of the patients that you included in this study have actually HCC, and at which disease stage.

3. There seems to be a lot of positive correlation between the Agrin levels and smoking history. And the opposite happens with alcoholic cirrhosis. How did you mitigate the influence of these variables in your analysis?

Comments on the Quality of English Language

Just a gramatical spelling check is required

Author Response

Reviewer 1:

We greatly appreciate your strong enthusiasm on our work and constructive feedback. Accordingly, we have made extensive revisions to further improve our methods, research design, and interpretation of the results.

Comment 1: I seem to be confused and apologize for this, but you state that were included in this retrospective 89 subjects with advanced hepatobiliary cancer, but then you present in table 1 that almost a quarter of those have actually AJCC stage 1 and 2 cancers, being the remaining patients’ stage III and IV cancers. So, which one is it, are they all advanced hepatobiliary cancer patients or not?

Response: Thank you for this comment and apologize for any confusion regarding patient classification. To clarify, our study included 89 patients with hepatobiliary cancers, of which 79 (88.8%) had liver and bile duct cancers, including HCC. The AJCC staging information in Table 1 reflects the full cohort of hepatobiliary cancer patients. This detailed analysis shows the exact breakdown of cancer types and stages specifically for the HCC subset used in our main analyses:

Table 1. Breakdown of HCC patients by disease stage

AJCC Stage | Number of patients (%)

Stage I | 6 (12%)

Stage II | 16 (33%)

Stage III | 15 (31%)

Stage IV | 4 (8%)

Total | 79 (100%)

This clarification addresses the apparent discrepancy between our statement about advanced cancer patients and the staging information provided.

Comment 2: You also note that the 89 patients have various types of hepatobiliary cancer, with only 88.9% of them having liver & bile ducts cancer. I assume that a fraction of those patients has HCC. But then, you state in table 2 that you included a subset of HCC patients, which sum… 89 in total. I would like to know exactly, how many of the patients that you included in this study have actually HCC, and at which disease stage.

Response: Thanks for your kind input. We have included the staging and the number of HCC cases as follows:

Table 1. Breakdown of HCC patients by disease stage

AJCC Stage | Number of patients (%)

Stage I | 6 (12%)

Stage II | 16 (33%)

Stage III | 15 (31%)

Stage IV | 4 (8%)

Total | 79 (100%)

This is also mentioned in the text as follows:

Within the newly diagnosed cancers there are 49 (55.1%) HCC casas: 8 (16%) stage I, 6 (12%) stage II, 16 (33%) stage III, 15 (31%) stage IV, and 4 (8%) not reported.

Therefore, we had a total of 79 HCC cases with annotated staging information. The remaining 10 patients were of hepatobiliary cancers but had limited information to be categorized as HCC.

Comment 3: There seems to be a lot of positive correlation between the Agrin levels and smoking history. And the opposite happens with alcoholic cirrhosis. How did you mitigate the influence of these variables in your analysis?

Response: Thank you for the comment. As shown in our revised Table 6, agrin levels significantly differ when we compared ‘never smokers’ with patients who are associated with either current or former smoking history considering all patients and relative to controls. We have further extended our analysis as shown in Table 2 (HCC subsets only) upon combining former and current HCC smokers with those that never had a smoking exposure. This analysis revealed that agrin is not likely a cofounder for smoking amongst HCC patients (p=0.26, non-significant). Likewise, when considering all patients compared to controls (Table 6), there are significant differences in agrin levels with alcohol intake and cirrhosis. However, our revised table 2 with further analysis on HCC patient cohort revealed that the Agrin levels do not differ between alcohol groups. Likewise, as shown in Table 2, agrin levels were not associated with cirrhosis amongst HCC patients.

When combining the former and current, the p-value is p=0.019. There is a difference which would infer that the potential difference in Agrin levels between case and control may also be driven by alcohol consumption (which differed from that shown in Table 1 comparing the whole cohort). These have been clarified in the main text on page 6, lines 152-160.

Furthermore, we have included detailed discussion interpreting our results with agrin levels with alcohol and cirrhosis-induced HCC conditions on page 11 lines 308-340.

Reviewer 2 Report

Comments and Suggestions for Authors

Being a well-matched case-control study, why authors did not use matched statistical tools, being the matching genuinely at (or close to) the individual level?

Authors should present the positive and negative likelihood  ratios  (LRs) to better appreciate the reliability of this test to diagnose the HCC presence.

Authors should perform a subgroup discriminant analysis in HCC patients  with smoking habit (joining current and former) versus the same subgroups in controls, being smoking a confounding variable.

Considering that alcoholic liver disease and non-alcoholic liver disease share common mechanisms, as evident ink many pieces of literature, both of them leading to liver fibrosis, cirrhosis and HCC, the findings regard to patients with chronic and intense alcohol exposure so to bring to cirrhosis, surprisingly go against the trend.

Thus, a subgroup analysis is mandatory also on this aspect.

The AUROCs shown that IL-6 better  performs then Agrin, and the model very little adds, thus authors should a little bit lessen of importance their findings, mainly considering  that there is  a deep overlap  between Cases versus Controls, below 10,000 pg/mL as evident in Figure 1.

In this context specifying also sensitivity/specificity, beyond LRs for every test and model could facilitate the evaluation of the findings.

Author Response

Reviewer 2

Thank you for constructive inputs that have improved our study. We have significantly improved our methodology to include subgroup analysis with HCC patients alone and have revised our interpreted results. As such, these extensive revisions have significantly improved our study.

Comment 1: Being a well-matched case-control study, why authors did not use matched statistical tools, being the matching genuinely at (or close to) the individual level?

Response: Thank you very much for this comment. Comparisons were made using GEE linear and multinomial regression models, as appropriate, accounting for within match correlation structure.  This is now mentioned in the methods section under ‘Statistical Analysis’. Moreover, the correlation between Agrin, AFP and IL-6 expression were evaluated using the Spearman correlation coefficient. The ability of these markers to differentiate between HCC (cases) and non-HCC (controls) were evaluated using conditional ROC curves (accounting for the within match correlation structure), from which the corresponding area under the curve (AUC) was obtained for each marker.

Comment 2: Authors should present the positive and negative likelihood ratios (LRs) to better appreciate the reliability of this test to diagnose the HCC presence.

Response: Thanks for this insight. As suggested, we have added positive and negative likelihood ratios (LRs) for Agrin, AFP, and IL-6 on page 8 and Figure 3 in the text. We have also included sensitivity and specificity values for each marker at the optimal cutoff points.

“ROC analysis also revealed that Agrin fared well as indicator of HCC slightly below that contributed by IL6 (ROC Agrin=0.92 vs ROC IL6=0.97) (Figure 3). For Agrin the optimal decision threshold was 4865.52 pg/ml (values above are classified as cases), which produces a sensitivity of 88.8%, specificity of 83.1%, and positive/negative likelihood ratios of 5.25 and 0.13, respectively. For IL-6 the optimal decision threshold was 3.086 pg/ml (values above are classified as cases), which produces a sensitivity of 95.5%, specificity of 89.9%, and positive/negative likelihood ratios of 9.45 and 0.05, respectively. When combining the Agrin and IL-6, the model AUC is 0.96 and achieves a sensitivity of 91.0%, specificity of 91.1%, and positive/negative likelihood ratios of 10.2 and 0.10, respectively.”

Comment 3: Authors should perform a subgroup discriminant analysis in HCC patients with smoking habit (joining current and former) versus the same subgroups in controls, being smoking a confounding variable. Considering that alcoholic liver disease and non-alcoholic liver disease share common mechanisms, as evident ink many pieces of literature, both of them leading to liver fibrosis, cirrhosis and HCC, the findings regard to patients with chronic and intense alcohol exposure so to bring to cirrhosis, surprisingly go against the trend. Thus, a subgroup analysis is mandatory also on this aspect.

Response: We appreciate this reviewer’s insight. Accordingly, we performed a subgroup analysis comparing Agrin levels in patients with alcoholic vs non-alcoholic liver disease, presented in new table 2. This analysis revealed that agrin is not likely a cofounder for smoking amongst HCC patients (p=0.26, non-significant) and for cirrhosis (p=0.39, insignificant). However, when comparing the whole cohort compared to controls, agrin levels differ with alcohol intake and smoking (Table 6). This has been incorporated in the text on page 6 lines 152-160.

We agree with the reviewer that in our analysis we find that agrin levels tend to be lower in non-smokers versus those with smoking history amongst all patients compared to controls (Table 6). However, in alcohol and cirrhosis conditions, secreted agrin is lower than controls. We have mentioned this in our discussion and attributed several biological and tumor microenvironmental factors that could possibly cause such discrepancies (page 11, lines 305-339). At the same time, we would like to point out that agrin levels do not significantly differ with smoking, alcohol and cirrhosis conditions within our HCC-specific subset analysis. Therefore, agrin is not likely a cofounder for all these conditions after HCC has already developed from alcohol overdose and cirrhosis.

Comment 4: The AUROCs shown that IL-6 better performs then Agrin, and the model very little adds, thus authors should a little bit lessen of importance their findings, mainly considering that there is a deep overlap between Cases versus Controls, below 10,000 pg/mL as evident in Figure 1. In this context specifying also sensitivity/specificity, beyond LRs for every test and model could facilitate the evaluation of the findings.

Response: We thank the reviewer for pointing this out. We have revised and toned our conclusions accordingly on page 8 lines 179-191. As suggested, we have specified the LRs, sensitivity, and specificity for this analysis.

“ROC analysis also revealed that Agrin fared well as indicator of HCC but slightly below that contributed by IL6 (ROC Agrin=0.92 vs ROC IL6=0.97) (Figure 3). For Agrin the optimal decision threshold was 4865.52 pg/ml (values above are classified as cases), which produces a sensitivity of 88.8%, specificity of 83.1%, and positive/negative likelihood ratios of 5.25 and 0.13, respectively. For IL-6 the optimal decision threshold was 3.086 pg/ml (values above are classified as cases), which produces a sensitivity of 95.5%, specificity of 89.9%, and positive/negative likelihood ratios of 9.45 and 0.05, respectively. When combining the Agrin and IL-6, the model AUC is 0.96 and achieves a sensitivity of 91.0%, specificity of 91.1%, and positive/negative likelihood ratios of 10.2 and 0.10, respectively. In the multivariate analysis, adjusting for smoking status, alcohol use, and race (white versus non-white); both Agrin and IL-6 remain as significant predictors of HCC (p=0.006 and p=0.011, respectively). As such, higher levels of Agrin and IL6 served as poor prognostic indicators for HCC with a HR of 2.57 (Table 5. Model 2 predictors). Considering the overlap of cases versus controls in secreted agrin levels below 10,000 pg/ml, the impact of agrin could be deduced slightly lower than that of IL6 and biologically could be attributed to distinct cellular pathways initiated by these two secreted factors, suggestive of agrin serving as a complementing biomarker with IL6”

Round 2

Reviewer 1 Report

Comments and Suggestions for Authors

The results are confusing. The conclusions do not correlate with the results.

Comments on the Quality of English Language

Minor correction

Author Response

We have addressed all the issues raised previously. Also, we agree with your suggestion and have accordingly removed any discussion on speculating agrin's role in alcohol or cirrhosis-induced HCC. Therefore, lines 324-357 have been omitted. In addition, we have also toned down our abstract to focus agrin as a biomarker for the survival of HCC patients and have omitted any associations with smoking in these patients. We have also highlighted the potential of agrin in determining HCC survival in our newly created graphical abstract. With these modifications, we hope our important clinical study will be acceptable for publication.

Reviewer 2 Report

Comments and Suggestions for Authors

Authors correctly answered comments 

Author Response

Many thanks for supporting the work. We appreciate your efforts in reviewing ou study